# FROM STATIC TO DYNAMIC: LEVERAGING IMPLICIT BEHAVIORAL MODELS TO FACILITATE TRANSITION IN OFFLINE-TO-ONLINE REINFORCEMENT LEARNING

## ABSTRACT

Transitioning reinforcement learning (RL) models from offline training environments to dynamic online settings faces critical challenges because of the distributional shift and the model inability in effectively adapting to new, unseen scenarios. This work proposes the **B**ehavior **A**daption **Q**-Learning (BAQ), a novel framework facilitating smoother transitions in offline-to-online RL. BAQ strategically leverages the implicit behavioral model to imitate and adapt behaviors of offline datasets, enabling the model to handle out-of-distribution state-action pairs more effectively during its online deployment. The key to our approach is the integration of a composite loss function that not only mimics the offline data-driven policy but also dynamically adjusts to new experiences encountered online. This dual-focus mechanism enhances the model's adaptability and robustness, reducing Q-value estimation errors and improving the overall learning efficiency. Extensive empirical evaluations demonstrate that BAQ significantly outperforms existing methods, achieving enhanced adaptability and reduced performance degradation in diverse RL settings. Our framework sets a new standard for offline-to-online RL, offering a robust solution for applications requiring reliable transitions from theoretical training to practical, real-world execution.

## 1 INTRODUCTION

Offline reinforcement learning (RL) has attracted impressive attention for its ability to learn policies from the offline static datasets without requiring direct interaction with the environment (Xie et al., 2021; Zhang & Zanette, 2024). This is particularly valuable in critical domains such as robotics Rafailov et al. (2023), navigation Zhao et al. (2023), and manipulations Rajeswaran et al. (2017), where collecting real-time data is either unsafe, impractical, or prohibitively expensive. However, deploying the models that are trained from the offline dataset in dynamic real-world environments poses severe challenges. Since the static datasets are generated by unknown behavior policies, they often lack crucial information on rare or unexplored states and actions (Fu et al., 2020). When the trained model is deployed in a real-world environment, it encounters the out-of-distribution (OOD) data (Yang et al., 2022), i.e., unseen state-action pairs, which have a different distribution from the offline dataset. This distributional mismatch between the OOD data and the offline dataset results in inaccurate Q-value estimates, which in turn misguide the agent's policy and result in learning performance degradation, a problem referred to as bootstrap error (An et al., 2021).

A variety of solutions have been proposed to rectify the Q- estimation caused by the bootstrap error (Bai et al., 2022). Among these, some methods impose conservative constraints during offline training to penalize the overestimation of Q-values for OOD actions. For instance, conservative Q-learning (CQL) (Kumar et al., 2020) trains pessimistic value functions (Wu et al., 2021; Blanchet et al., 2024) that inherently bias the agent towards more conservative actions, thereby reducing the risk of overestimation. However, these conservative methods impede the learning process caused by excessively restricting the policy, which limits the agent's ability to explore and refine the initial offline policy. Other methods prioritize the inclusion of online samples during fine-tuning, thereby allowing the agent to adjust its Q-value estimates based on more current and relevant experiences (Wu et al., 2019; Lee et al., 2021). Aiming to balance replay buffers, these methods help the agent move beyond the constraints of the offline dataset. In addition, behavior regularization methods constrain

the policy to remain close to the behavior policy used during offline training in order to mitigate the impact of OOD data (Ran et al., 2023). However, many of these approaches are sensitive to hyperparameter tuning, which require careful tuning to achieve the right balance between leveraging offline data and adapting to online samples. Additionally, the methods relying on estimating density ratios or measuring distributional divergences between offline and online data are resource-intensive and difficult to implement (Peng et al., 2023; Li et al., 2022). These challenges highlight the need for more robust, scalable, and computationally efficient approaches to managing distribution shift in offline-to-online RL, which would acheive accurate Q-value estimation and stable policy updates during the transition to online fine-tuning(Prudencio et al., 2023).

This work introduces a novel framework for enhancing online fine-tuning in offline-to-online RL by leveraging a behavior cloning model trained on offline datasets. Our approach is specifically designed to facilitate the transition from offline-trained models to dynamic environments, enabling agents to quickly adapt to new conditions without performance degradation. Below we outline the significant contributions of our work:

1. **Behavior Cloning Integration:** We deploy a behavior cloning model that serves as a foundational reference for adapting to the new data encountered in online fine-tuning. This model is instrumental in predicting and adjusting to discrepancies between behaviors of offline dataset and actual online interactions, thus smoothing the offline-to-online transition.

2. **Dynamic Q-value Adjustment:** Our methodology introduces a modification to the loss function that uses insights from the behavior cloning model to dynamically adjust Q-value estimations. By computing a weighting factor that diminishes the impact of novel state-action pairs, we mitigate the risk of significant Q-estimation errors with OOD data.

3. **Priority-Based Sample Rebalancing:** We refine the replay buffer strategy through a priority sampling strategy, where sample priorities are dynamically adjusted based on their deviation from the behavior cloning model's predictions. This strategy effectively biases training towards transitions more aligned with the current policy.

4. **Empirical Validation and Performance Gains:** Extensive empirical analyses demonstrate that our framework significantly outperforms existing methods in offline-to-online RL. These results highlight the practical benefits of our contributions from simulated or theoretical training environments to real-world conditions.

These contributions systematically address the limitations inherent in traditional offline-to-online learning transitions and set a new benchmark for the field, offering methodologies that can be directly applied or adapted for a wide range of practical reinforcement learning applications.

## 2 RELATED WORK

Offline-to-online reinforcement learning (RL) has gained increasing attention as it allows models trained on static datasets to be fine-tuned through dynamic, real-time interactions. The transition from offline to online presents several challenges, including Q-value estimation errors (Ghasemipour et al., 2021), distributional shift (Qi et al., 2022), and efficient sampling strategies (Guo et al., 2023). Below, we provide a structured overview of the existing methods seeking to address these issues.

**Reducing Q-Value Estimation.** A major challenge in offline-to-online RL is the accurate Q-value estimation, especially when agents encounter OOD data during online fine-tuning. Q-value estimation errors can lead to suboptimal policy updates, limiting the agent's performance. Both SO2 (Zhang et al., 2024) and SUF (Feng et al., 2024) address this issue via reducing bias in Q-value estimation during the online training phase. SO2 introduces a perturbed value update method to smooth out biased Q-values and prevent premature exploitation of suboptimal actions. Similarly, SUF manages Q-value estimation by adjusting the Update-to-Data (UTD) ratio, which helps prevent overfitting to the offline dataset and allows the agent to explore more effectively during fine-tuning. Meanwhile, Cal-QL (Nakamoto et al., 2024) and FamO2O (Wang et al., 2024) adopt a more adaptive approach to Q-value estimation by calibrating the Q-values during the offline phase and progressively updating them as the agent encounters new data online.

**Managing Distributional Shift.** Managing distributional shift is crucial in offline-to-online RL, as offline-trained policies can struggle to generalize when faced with novel online data. Methods

like GCQL (Zheng et al., 2023) and Off2On (Lee et al., 2022) balance conservative offline training with more exploratory updates during online fine-tuning. GCQL adopts greedy update strategy to adapt to new data, while Off2On uses a balanced replay buffer to prioritize near-on-policy samples. The method in Ball et al. (2023) deploys the Layer Normalization (LayerNorm) to prevent the over-extrapolation during online interactions. Along with symmetric sampling, it improves policy stability and performance. Additionally, PEX (Yu & Zhang, 2023) mitigates distributional shifts by retaining the offline policy while adapting a new policy to the online environment. These methods emphasize balancing conservatism and exploration to manage distribution shifts effectively.

**Issues in Existing Works.** Existing offline-to-online RL methods fail to directly address the key challenges during the transition phase. Rather than using the behavior of the offline data to directly guide the online policy, they often rely on indirect mechanisms such as imposing constraints, conservative updates, or introducing additional measurements like bias correction or pessimistic value estimates. These methods slow down learning and lead to instability during fine-tuning as well. A more direct approach, free from such adjustments, would allow smoother transitions and faster learning.

## 3 PRELIMINARIES

### 3.1 REINFORCEMENT LEARNING

RL is a framework in which an agent learns to maximize cumulative rewards by interacting with an environment (Ernst & Louette, 2024; Mahadevan, 1996). The problem is often modeled as a Markov Decision Process (MDP), defined by a tuple $(\mathcal{S}, \mathcal{A}, P, R, \gamma)$, where $\mathcal{S}$ is the state space, $\mathcal{A}$ is the action space, $P(s'|s, a)$ is the transition probability, $R(s, a)$ is the reward function, and $\gamma \in [0, 1)$ is the discount factor. At each timestep $t$, the agent observes a state $s_t$, takes an action $a_t \in \mathcal{A}$, receives a reward $r_t = R(s_t, a_t)$, and then transitions to a new state $s_{t+1}$ according to $P(s_{t+1}|s_t, a_t)$. The goal in RL is to find a policy $\pi(a|s)$ that maximizes the expected cumulative return:

$$\pi^* = \arg\max_\pi \mathbb{E}_\pi \left[ \sum_{t=0}^{\infty} \gamma^t R(s_t, a_t) \right]. \tag{1}$$

As for offline RL, the agent learns exclusively from a static dataset $\mathcal{D} = \{(s, a, r, s')\}$ that is collected by a behavior policy $\mu$, without further interaction with the environment. The primary challenge in offline RL is that the dataset $\mathcal{D}$ typically has limited coverage of the state-action space, leading to Q-function overestimation for OOD actions. This overestimation can result in suboptimal policies when deployed online. After offline training process, offline-to-online RL extends offline learning by allowing the agent to fine-tune its policy through limited online interaction. During the fine-tuning phase, the agent is expected to balance the knowledge learned from the offline dataset with new experiences from the online phase, adapting the policy without overfitting to OOD actions or destabilizing the learning process. Ensuring stability during this transition is crucial for the success of offline-to-online RL.

### 3.2 BEHAVIORAL CLONING

Behavioral Cloning (BC) (Torabi et al., 2018) is a supervised learning method used to train an agent to imitate the actions demonstrated by an expert or recorded in a dataset. The goal of BC is to directly learn a policy $\pi_\theta(a|s)$ that predicts actions $a$ given states $s$ by minimizing the error between the predicted actions and the expert actions. The loss function for BC is typically defined as the negative log-likelihood of the expert actions under the learned policy:

$$\mathcal{L}_{\text{BC}} = \mathbb{E}_{(s,a)\sim\mathcal{D}} \left[ -\log \pi(a|s) \right], \tag{2}$$

where $(s, a) \sim \mathcal{D}$ denotes the state-action pairs $(s, a)$ sampled from the dataset $\mathcal{D}$, and $\pi(a|s)$ represents the probability that the policy $\pi$ takes action $a$ in state $s$. The objective is to maximize the likelihood of taking the expert's actions in the given states.

### 3.3 CONSERVATIVE Q-LEARNING

Conservative Q-Learning (CQL) (Kumar et al., 2020) is an offline RL algorithm designed to address the overestimation problem caused by OOD actions in the dataset. CQL modifies the Q-value updates by regularizing the policy towards actions seen inside the dataset and penalizing Q-values for actions outside the dataset.

CQL minimizes the following loss function, which penalizes large Q-values for unseen actions:

$$\mathcal{L}_{\text{CQL}}(Q) = \alpha \cdot \mathbb{E}_{s\sim\mathcal{D}}\left[\log \sum_{a'} \exp(Q(s,a')) - Q(s,a)\right] + \frac{1}{2}\cdot\mathbb{E}_{(s,a,s')\sim\mathcal{D}}\left[\left(Q(s,a) - \hat{\mathcal{B}}^{\pi}\hat{Q}_{\text{target}}(s,a)\right)^2\right]. \quad (3)$$

Here, $\alpha$ is a hyperparameter controlling the degree of conservatism. The loss function has two main terms. The first term encourages the Q-values of actions $a$ from the dataset $\mathcal{D}$ to be higher than those for other actions $a'$ (potentially sampled from a broader action space), thus penalizing the Q-values for OOD actions and reducing overestimation. The second term is a standard temporal difference (TD) loss, which aligns the Q-values with the target Q-values, promoting accurate estimation of actions in the dataset. Minimizing this loss enables CQL to obtain conservative Q-value estimates for actions that are insufficiently represented in the offline dataset, for which the overestimation risks are effectively mitigated.

### 3.4 IMPLICIT Q-LEARNING

Implicit Q-Learning (IQL) (Kostrikov et al., 2021) is an offline RL algorithm designed to address the challenge of overestimating Q-values for OOD actions without explicitly querying these unseen actions. IQL achieves this by leveraging *expectile regression*, which enabling the algorithm to prioritize actions that are well-supported by the offline dataset.

The $\tau$-expectile provides a flexible way to balance between mean-based estimation ($\tau = 0.5$) and maximizing Q-values ($\tau \to 1$). The value function is learned by minimizing the following expectile regression loss:

$$\mathcal{L}_{IQL}(V) = \mathbb{E}_{(s,a)\sim\mathcal{D}}\left[L_\tau^2\left(Q(s,a) - V(s)\right)\right], \quad (4)$$

where $L_\tau^2(u) = |\tau - \mathbb{1}(u < 0)|u^2$. Once the value function is learned, the Q-function is updated by minimizing the mean squared error loss between the Q-values and the expected returns, incorporating the learned value function to handle stochastic transitions in the environment. The Q-function update is expressed as:

$$\mathcal{L}_{IQL}(Q) = \mathbb{E}_{(s,a,s')\sim\mathcal{D}}\left[\left(r(s,a) + \gamma V(s') - Q(s,a)\right)^2\right]. \quad (5)$$

IQL extracts the policy through advantage-weighted behavioral cloning, where the learned Q-function is deployed to prioritize actions with higher advantages. The final policy maximizes Q-values while maintaining proximity to the behavior policy from the offline dataset. It prevents divergence from the data for stable performance in offline settings. By alternating between expectile regression and Q-function updates, IQL efficiently performs multi-step dynamic programming, resulting in robust policy performance.

## 4 METHODOLOGY

### 4.1 THE OOD NATURE

Offline-to-online RL offers significant advantages by allowing agents to be pre-trained on static datasets, reducing the amount of costly and time-consuming online interactions required. However, a major challenge arises when transitioning from offline training to online fine-tuning: the state-action pairs encountered during the online phase often differ substantially from those in the offline dataset. This results in a significant distribution shift, introducing a large amount of OOD data into the agent's experience buffer. The presence of OOD data can lead to inaccurate Q-value estimations, which in turn destabilizes the learning process. As the agent attempts to adapt to the new environment, these errors in Q-value estimation can misguide policy updates, leading to performance degradation and slower learning progress. Addressing this challenge is critical for efficient and stable fine-tuning in offline-to-online RL.

To demonstrate this, we re-sample actions using the well-trained offline model, based on states from the offline dataset. As shown in Fig. 1a and 1b, even when the model is exposed to states identical to those in the offline dataset, the actions it produces often deviate from the corresponding actions in the dataset. This deviation highlights the OOD nature of the model's behavior, which arises when the model begins interacting with the environment and generating actions on its own. To tackle this challenge, accurately representing the offline data distribution is essential, as it provides a stable reference for the agent during its transition to online interactions. This motivates our exploration of behavior cloning (BC) as a potential solution. By replicating the policy that generated the offline dataset, BC helps keep the agent's actions aligned with expert behavior. As depicted in Fig. 1c and 1d, BC significantly reduces the mean squared error (MSE) between the model's predicted

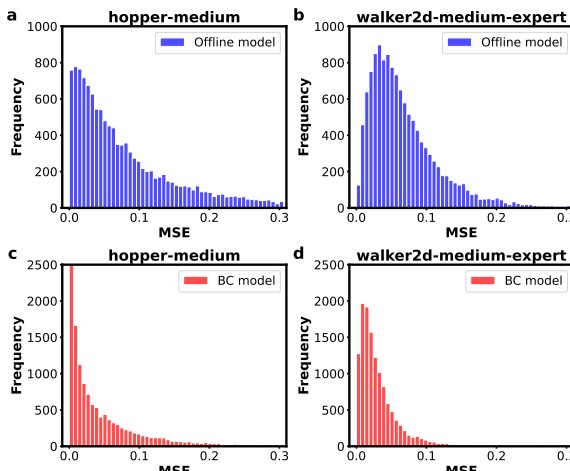

Figure 1: Comparison between the actions generated by the model and those in the offline dataset. (a, b) show the results by the offline-trained model. (c, d) show the results by the BC model trained on the offline dataset.

actions and the offline dataset, demonstrating its effectiveness in stabilizing the learning process during the offline-to-online transition and thus mitigating the OOD issue.

## 4.2 REDUCING Q-VALUE ESTIMATION BIAS WITH WEIGHTED Q-LEARNING

We begin by training a behavior cloning (BC) model, $\pi_{BC}$, on an offline dataset, which serves as the reference policy. During online fine-tuning, a second policy, $\pi_{on}$ started from offline training process, interacts with the environment and collects new state-action pairs. These interactions, along with the offline data, are stored in a replay buffer. Since $\pi_{on}$ encounters the OOD data that are not well-covered by the offline dataset, this could lead to inaccurate Q-value estimates. To address this, we introduce a weighting mechanism that adjusts the Q-value updates based on the alignment between $\pi_{BC}$ and $\pi_{on}$.

To ensure stable learning, we define a distance measure that quantifies the divergence between the actions predicted by the offline policy $\pi_{BC}$ and the actions stored in the replay buffer, which consists of data from both offline and online interactions. both offline and online interactions. The weight $w(s, a)$ for each state-action pair $(s, a)$ is given by:

$$w(s, a) = \exp\left(-\frac{\text{mean}\left((\pi_{BC}(s) - a)^2\right)}{k_q}\right).$$
(6)

Here, the parameter $k_q$ is a scalar that controls the sensitivity of the weight to the action differences. This weight penalizes large discrepancies between the actions of $\pi_{BC}$ and the observed actions, guaranteeing that the fine-tuning process gives more importance to regions of the state space where the policies are more aligned. We incorporate the distance measure $w(s, a)$ into the Q-loss functions of both CQL and IQL to stabilize Q-value updates during fine-tuning.

**with CQL:** To incorporate the weighting mechanism, we modify the conservative penalty term of CQL in Eq. 3 as follows:

$$\frac{1}{2} \cdot \mathbb{E}_{(s,a,s') \sim \mathcal{D}}\left[w(s, a) \cdot \left(Q(s, a) - \hat{\mathcal{B}}^\pi \hat{Q}_{\text{target}}(s, a)\right)^2\right].$$
(7)

Eq. 7 addresses Q-value bias during online fine-tuning by weighting updates based on the similarity between actions from the offline-trained policy $\pi_{BC}$ and the replay buffer, which contains both offline and online data. The weight $w(s, a)$ reduces the impact of OOD data, enabling the updates

to concentrate state-action pairs aligned with $\pi_{BC}$. This helps mitigate Q-value overestimation and stabilizes learning.

**with IQL:** IQL first learns a value function, and then updates the Q-function using the learned value function to handle stochastic transitions in the environment. On the basis of Eq. 4, the value function is learned by minimizing the following weighted loss:

$$\mathcal{L}_{\text{IQL}}(V) = \mathbb{E}_{(s,a)\sim\mathcal{D}} \left[ w(s,a) \cdot L_\tau^2 \left( Q(s,a) - V(s) \right) \right]. \tag{8}$$

The weight $w(s,a)$ adjusts the importance of each state-action pair, which concentrates more on state-action pairs where the online policy aligns with the policy $\pi_{BC}$. Once the value function is learned, the Q-function is updated by minimizing the following weighted loss:

$$\mathcal{L}_{\text{IQL}}(Q) = \mathbb{E}_{(s,a,s')\sim\mathcal{D}} \left[ w(s,a) \cdot \left( r(s,a) + \gamma V(s') - Q(s,a) \right)^2 \right]. \tag{9}$$

By incorporating the weight $w(s,a)$ into both the value and Q-function updates, the learning process emphasizes regions of the state-action space where the online policy is more closely aligned with the behavior of the offline data.

### 4.3 BC-DIVERGENCE PRIORITY SAMPLING FOR FINE-TUNING

The new data collected through interactions with the environment offer valuable insights that are absent from the offline dataset, which captures previously unseen states and actions. This fresh information is essential for fine-tuning the online policy, as it helps the agent adapt to novel situations more effectively. Given this insight, we design a priority sampling mechanism for the replay buffer that optimally balances learning from both the most informative new data and the critical offline data, ensuring efficient adaptation and policy improvement. The priority of a new transition $(s, a, s', r)$ is calculated as:

$$\rho = \left( \frac{\|\pi_{BC}(s) - a\|}{k_\rho} + 1 \right)^\alpha, \tag{10}$$

where $k_\rho$ is a normalization constant that controls the scale of the divergence, and $\alpha$ is a hyperparameter that controls the sensitivity of the priority to the action differences. The sampling probability of each transition $(s, a, s', r)_i$ in the replay buffer is then determined as: $\mathbb{P}_{(s,a,s',r)_i} = \frac{\rho_i}{\sum_j \rho_j}$.

Our proposed mechanism prioritizes transitions where the online policy deviates significantly from the behavior cloning policy. By focusing on these transitions, the model can better learn from the new state-action pairs it encounters during online interactions.

### 4.4 PRACTICAL ALGORITHM

The Algorithm 1 outlines the steps for our proposed Behavior Adaption Q-Learning (BAQ). BAQ starts by training a BC model $\pi_{BC}$ on an offline dataset, followed by the priority sampling and weighted Q-learning during the online fine-tuning phase. The priority sampling guarantees that the replay buffer emphasizes transitions most beneficial for policy improvement, while the weighted Q-learning updates refine Q-value estimates based on the alignment between $\pi_{BC}$ and the online policy $\pi_{on}$.

## 5 EXPERIMENTS

### 5.1 ENVIRONMENTS SETUP

**Evaluation.** We evaluate BAQ on MuJoCo (Todorov et al., 2012) tasks from the D4RL-v2 dataset[1], which includes three environments: HalfCheetah, Walker2d, and Hopper. Each environment contains datasets collected by policies of varying quality, categorized as Medium, Medium-Replay, and Medium-Expert. We report the performance on the standard normalized scores in D4RL, averaged over 4 seeds.

---

[1]https://github.com/Farama-Foundation/D4RL

---

**Algorithm 1** Behavior Adaption Q-Learning (BAQ)

---

1: **Initialize:** Behavior cloning policy $\pi_{BC}$ from offline dataset
2: **Initialize:** Offline policy $\pi_{off}$ as $\pi_{on}$ and associated Q networks from offline training process
3: **Initialize:** Replay buffer $\mathcal{D}$ with offline data, set priority $\rho = 1$ for all offline transitions
4: **for** each iteration **do**
5:     Interact with environment using $\pi_{on}$ and collect new transitions
6:     Store new interactions in $\mathcal{D}$ with priority $\rho$ in Eq. 10
7:     Sample batch $\{(s, a, r, s', \rho)\}$ from $\mathcal{D}$ using priority sampling
8:     Compute weight $w(s, a)$ for transitions in batch using Eq. 6.
9:     Update the Q-functions using $\mathcal{L}_{\text{CQL}}$ in Eq. 7 or $\mathcal{L}_{\text{IQL}}$ in Eq. 8 and Eq. 9
10:    Update policy $\pi_{on}$
11: **end for**

---

**Setup.** We train the BC policy $\pi_{BC}$ for 1 million steps with a learning rate of $3 \times 10^{-4}$. Our algorithm is built upon the FamO2O framework, with both offline agents, CQL and IQL, implemented in the JAX version[2]. All models are pre-trained for 1 million steps using a learning rate of $3 \times 10^{-4}$, maintaining consistency throughout the training process. For the hyperparameters used in BAQ, we set $(k_q = 1, k_\rho = 2)$ for larger datasets (Medium-Expert) and $(k_q = 2, k_\rho = 1)$ for smaller datasets (Medium-Replay) in both IQL and CQL. For the Medium dataset, we use $(k_q = 2, k_\rho = 0.5)$ in CQL and $(k_q = 0.5, k_\rho = 0.5)$ in IQL. More implementation details, including specific hyperparameter settings, can be found in the Appendix.

**Comparison.** We evaluate the following baselines, starting with offline-trained models and applying specific techniques during the online fine-tuning phase:

- **IQL Kostrikov et al. (2021)**: A value-based RL method that learns from offline data without explicitly estimating the behavior policy, utilizing expectile loss to strike a balance between over- and underestimation.

- **CQL Kumar et al. (2020)**: A pessimistic offline RL method that penalizes overestimation of OOD actions, promoting stability during offline-to-online transitions.

- **SO2 Zhang et al. (2024)**: A method that smooths biased Q-value estimates, preventing the exploitation of suboptimal actions during fine-tuning. SO2 is applied to both IQL and CQL. Further details are provided in the Appendix.

- **SUF Feng et al. (2024)**: A method that stabilizes fine-tuning by adjusting update ratios, thereby preventing policy collapse. SUF is applied to both IQL and CQL. Additional details are available in the Appendix.

- **Off2On Lee et al. (2022)**: A method that employs balanced replay and a pessimistic Q-ensemble to stabilize fine-tuning, mitigating distribution shift.

For fair comparison, none of the baselines use an ensemble strategy. Advanced methods such as FamO2O are excluded because they require access to the offline training phase for constructing policy families or calibrating Q-values. Since our setup begins with pre-trained offline models, these methods are not applicable during the fine-tuning phase.

## 5.2 MAIN RESULTS

As shown in Tab. 1, we evaluate IQL and CQL, along with their various extensions, across several locomotion tasks. The results highlight the effectiveness of our proposed method in both IQL and CQL settings. For the IQL experiments, although IQL+SO2 perform best on Hopper-Medium-Expert (94.6), our method demonstrates competitive results across most tasks, particularly in HalfCheetah-Medium-Expert, where it achieves 78.5, outperforming other IQL variants. In the CQL comparisons, our method shows significant improvements again, achieving much higher scores than the base CQL algorithm. Overall, our approach not only surpasses all baseline methods in total scores for both IQL and CQL but also demonstrates strong stability and generalization across diverse tasks. This under-

---

[2]https://github.com/LeapLabTHU/FamO2O

| | IQL | IQL + SO2 | IQL + SUF | IQL + Ours | CQL | CQL + SO2 | CQL + SUF | CQL + Ours |
|---|---|---|---|---|---|---|---|---|
| halfcheetah-me | 85.1 ±6.3 | 81.2 ±7.8 | 75.2 ±13.1 | 78.5 ±11.2 | 57.0 ±19.0 | 39.5 ±7.7 | 40.2 ±8.8 | 75.3 ±11.4 |
| hopper-me | 61.9 ±45.0 | 94.6 ±23.0 | 92.9 ±19.3 | 88.7 ±28.3 | 93.3 ±23.0 | 59.8 ±28.2 | 43.4 ±23.0 | 102.3 ±14.7 |
| walker2d-me | 109.6 ±1.4 | 107.9 ±3.2 | 108.7 ±1.0 | 109.9 ±1.8 | 110.1 ±0.6 | 101.6 ±11.2 | 89.6 ±19.1 | 109.3 ±0.99 |
| halfcheetah-mr | 44.8 ±1.3 | 44.7 ±0.6 | 43.4 ±2.5 | 44.7 ±0.7 | 47.8 ±0.4 | 46.5 ±0.9 | 45.9 ±1.2 | 46.0 ±0.95 |
| hopper-mr | 83.4 ±15.3 | 52.7 ±15.9 | 47.0 ±6.43 | 86.1 ±19.0 | 95.3 ±2.5 | 84.8 ±15.8 | 79.1 ±22.4 | 92.8 ±9.5 |
| walker2d-mr | 66.2 ±14.8 | 60.9 ±8.2 | 63.6 ±13.9 | 76.3 ±10.5 | 82.3 ±4.4 | 69.4 ±9.5 | 54.3 ±15.8 | 76.2 ±6.0 |
| halfcheetah-m | 47.7 ±0.5 | 46.1 ±0.4 | 45.9 ±0.4 | 47.7 ±0.3 | 48.6 ±0.5 | 46.8 ±0.6 | 45.6 ±0.8 | 48.4 ±0.46 |
| hopper-m | 64.5 ±9.7 | 55.8 ±5.9 | 51.5 ±4.5 | 70.6 ±8.4 | 70.8 ±4.2 | 57.1 ±10.1 | 58.2 ±11.6 | 68.9 ±8.48 |
| walker2d-m | 80.4 ±6.0 | 74.3 ±8.7 | 71.6 ±7.7 | 83.9 ±3.1 | 82.7 ±0.7 | 59.0 ±13.7 | 69.6 ±9.0 | 82.4 ±1.63 |
| Total | 643.44 | 618.27 | 599.71 | **686.26** | 687.92 | 564.61 | 525.95 | **702.32** |

Table 1: Performance comparison of IQL and CQL variants during the initial 30,000 steps of online fine-tuning across different datasets. me = medium-expert, mr = medium-replay, and m = medium.

scores the robustness and efficiency of our method in handling the challenges of offline-to-online RL.

In addition, both SO2 and SUF struggle significantly during the initial stages of online training. Although these methods show some advantages over the original CQL or IQL algorithms in later stages, they perform poorly without the cooperation of advanced techniques like the Q-ensemble, which helps mitigate issues such as overestimation and instability. This highlights a limitation of these approaches when used independently. In contrast, our method has a robust performance from the beginning without relying on such additional mechanisms. We emphasize that our comparisons are made fairly against the original algorithms. Our concentration on the core idea without incorporating extra optimizations guarantees a more direct and valid evaluation of performance improvements.

### 5.3 BC-Divergence Priority Sampling

The Tab. 2 presents a comparison between CQL, OFF2ON, and our method's Our-S variant, one of the two key components in our approach. While Our-S demonstrates improvements in certain tasks, such as HalfCheetah-Medium-Expert (65.02) and Hopper-Medium-Expert (97.61), the performance gains are not overwhelming. In tasks like Walker2d-Medium-Replay and HalfCheetah-Medium-Replay, Our-S performs slightly below CQL and OFF2ON. Despite these mixed results, Our-S still achieves the highest total score of 690.74, slightly outperforming CQL (687.92) and OFF2ON (688.83). This suggests that: while our sampling strategy has advantages in certain scenarios, its improvements are incremental rather than dramatic, contributing to a balanced enhancement across various tasks rather than a dominant performance.

| Dataset | CQL | OFF2ON | Our-S |
|---|---|---|---|
| halfcheetah-me | 56.95 | 59.95 | 65.02 |
| hopper-me | 93.29 | 94.84 | 97.61 |
| walker2d-me | 110.10 | 109.65 | 109.52 |
| halfcheetah-mr | 47.83 | 47.20 | 46.56 |
| hopper-mr | 95.32 | 96.54 | 97.64 |
| walker2d-mr | 82.29 | 82.33 | 76.06 |
| halfcheetah-m | 48.63 | 48.47 | 48.24 |
| hopper-m | 70.83 | 68.14 | 68.48 |
| walker2d-m | 82.69 | 81.72 | 81.61 |
| Total | 687.92 | 688.83 | **690.74** |

Table 2: Performance comparison of CQL, OFF2ON, and Our-S across different tasks.

### 5.4 Details in Training Process

Fig. 2 shows a comparison between IQL, SO2, SUF, and BAQ, revealing a nuanced performance across various tasks. Our BAQ consistently leads in most tasks with higher normalized scores early in the fine-tuning process. Notably, BAQ demonstrates a strong response right from the beginning, consistently outperforming other methods in the initial stages. In contrast, SO2 and SUF exhibit more of a struggle during the early training phase. In summary, our BAQ method proves to be highly effective, particularly in the early stages of fine-tuning.

### 5.5 Ablation Study

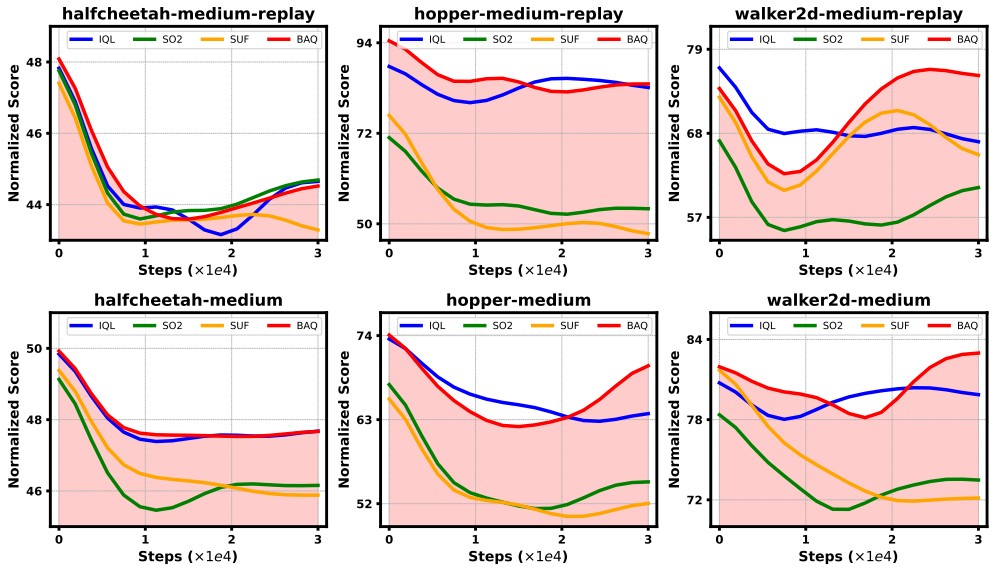

Figure 2: Training processes comparison of IQL, SO2, SUF, and our BAQ across various tasks.

The ablation study in Fig. 3 illustrates the impact of removing key components, Our-Q and Our-S, from both CQL and IQL. The results show noticeable performance degradation across all datasets, particularly in the Medium-Expert (me) and Medium-Replay (mr) settings, with the difference from our full method ranging from -5 to -30 normalized score points. Additionally, as the weighting term $w(s, a)$ in the Q loss function slows down the update progress, the performance of Our-Q tends to be lower than that of Our-S. This is reflected in the results where removing Our-Q causes a more significant performance drop compared to Our-S. Overall, the results highlight the critical role both components play in maintaining the strong performance of our full method, particularly in enhancing the stability and efficiency of the fine-tuning process in offline-to-online RL.

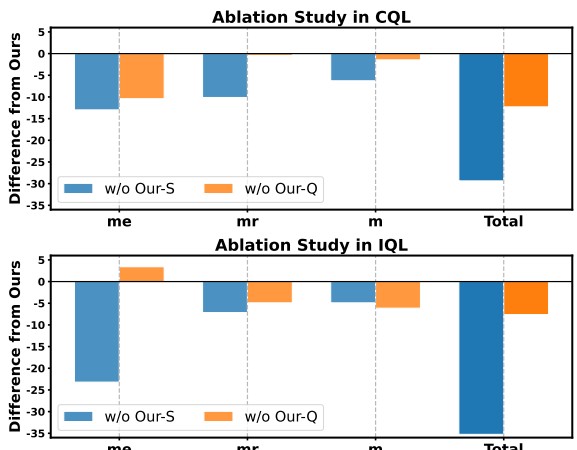

Figure 3: Ablation results for showing the performance drop when removing key components.

## 6 CONCLUSION

In this paper, we innovate Behavior Adaption Q-Learning (BAQ), a framework designed to facilitate smooth transitions from offline to online RL by integrating behavioral cloning and dynamic Q-value adjustment. Our prososed weighted loss functions and priority sampling address the issues of Q-value overestimation and distribution shift, respectively. Extensive experiments demonstrate that BAQ outperforms baseline methods such as IQL, CQL, SO2, and SUF. While BAQ shows robust performance and potential for broader applications, its effectiveness is limited by the size of the offline dataset, which can impact its ability to generalize during online fine-tuning. Future work could investigate strategies to reduce this dependency on large datasets, potentially through data-efficient learning techniques. Additionally, extending BAQ to more complex real-world environments could further validate its applicability and scalability.

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
