# OpenReview forum: "From Static to Dynamic: Leveraging Implicit Behavioral Models to Facilitate Transition in Offline-to-Online Reinforcement Learning"
_ICLR.cc/2025/Conference — Submitted to ICLR 2025_

### Official Review · Reviewer_u7BN · 2024-10-17

**Soundness:** 1
**Presentation:** 2
**Contribution:** 1
**Rating:** 3
**Confidence:** 5

**Summary:**

This paper presents a method called Behavior Adaption Q-Learning (BAQ) to solve the problem in offline-to-online RL setting, namely the distribution mismatch between the offline dataset and the online interaction. BAQ is instantiated as two parts, namely weighting the Q-learning process in CQL / IQL and adjusting the sample priority of the transition in the replay buffer. Some experiments demonstrate the performance of BAQ.

**Strengths:**

1. drawing from the innate problem of offline-to-online RL, this paper presents a simple method to improve the performance.
2. the pseudo-code of the Algorithm is clear.
3. the source code is provided

**Weaknesses:**

In my view, this paper is rather rudimentary, as the following weakness is obvious:
1. In the abstract, the authors claim that BAQ could reduce Q-value estimation errors and improve overall learning efficiency. However, their proposed method does not have any theoretical evidence to show how much learning efficiency can be improved or how many estimation errors can be reduced. If taking this as an empirical paper, then empirical contributions in and off BAQ should be strong enough, which seems to not be the case in the submission e.g. the results are only averaged by 4 seeds, and the experimental results are limited in MuJoCo.
2. The proposed method BAQ is simple and straightforward. However, the details of why it works are missing, adding more interpretation on both theoretical and experimental would be appreciated.
3. Imprecise Terminology: the bootstrap error does not only mean inaccurate Q-value estimates. Please refer to [1] for some details that the bootstrap error also covers the Bellman backup process, namely error accumulating. Only referring to the misguide but not referring to the bootstrapping process is imprecise.
4. Lack of references: the method [2] is also free from constraining the policy shift and balancing the sample replay based on the estimation of distribution divergence or density ratio. It would be appreciated if the authors could take it into discussion and comparison.
5. The OOD NATURE in Section 4.1 is unnecessary as this problem is not proposed the first time. Some similar interpretations can be found in [2].
6. "Advanced methods such as FamO2O are excluded ...". This is not an excuse for excluding the comparison as all the methods have access to the offline training dataset. If having the offline training datasets, the accessibility to the offline training phase is natural.
7. In Section 5.3, it seems that the BC-divergence priority sampling is of little use as the relative performance improvement is limited, e.g. only 1 or 2.
8. The experiments are averaged by multiple seeds, hence please report the deviation in Table 2, Figure 2 and Figure 3.
9. In Fig. 2, it seems that all the methods suffer a big performance drop during the online fine-tuning stage but why? As far as I know, the online fine-tuning stage should improve the asymptotic performance.

[1] Stabilizing Off-Policy Q-Learning via Bootstrapping Error Reduction. Aviral Kumar. et al
[2] Actor-Critic Alignment for Offline-to-Online Reinforcement Learning. Zishun Yu & Xinhua Zhang

**Questions:**

1. Why the topic is "implicit behavioral models"? As the paper proposed BAQ is instantiated on both CQL and IQL, how does the implicit show on the CQL variants?
2. Does the authors do parameter sweeping for all the baselines? The authors claim that BAQ significantly outperforms existing methods, providing some detailed information for parameter-tuning and sweeping could bring more confidence to the experiments.

**Details Of Ethics Concerns:**

NIL

---

### Official Review · Reviewer_XbsZ · 2024-10-30

**Soundness:** 2
**Presentation:** 1
**Contribution:** 2
**Rating:** 3
**Confidence:** 5

**Summary:**

The paper proposes a new offline-to-online method that uses heuristics to guide the online learning to focus on transitions that not OOD (with respect to the offline behaviors). In particular, the method works by weighing the bellman loss as well as adjusting how often the transitions are being sampled from the replay buffer based on how OOD the transitions are. Such technique can be readily applied to existing offline RL methods such as IQL and CQL and improve the fine-tuning sample efficiency on the D4RL benchmark.

**Strengths:**

- The proposed method is simple and described with great clarity.
- Empirical evaluations show improvements over prior techniques that improve offline-to-online RL methods.

**Weaknesses:**

*Misattribution and citation errors*
- L32-33 discusses about offline RL, but both of the cited works for this discussion are not focusing purely offline RL. (Zhang & Zanette, 2024) focuses on the setting where additional online data can be collected (informed by the offline data) and (Xie et al, 2021) focuses on offline-to-online RL.
- L34-35 discusses about benchmarks, but two of three papers cited did not first introduce the benchmarks they tested on (e.g., Zhao et al. (2023) and Rafailov et al. (2023). These papers are also about offline-to-online RL which is not suitable in the context of the offline RL discussion.

*Poorly justified claims*
- L239-240 “As depicted in …, demonstrating its effectiveness in stabilizing the learning process during the offline-to-online transition and thus mitigating the OOD issue” — The Figure referenced shows nothing about how the learning process is being stabilized. The only thing that the figure shows is the difference (MSE) between the model prediction and the offline data is reduced when the model is a BC model. This is also not surprising given that the BC model is directly being optimized to minimize the prediction error on the offline data.

*Missing important baselines comparisons.*
While the proposed technique helps existing offline RL methods to fine-tune better, the authors seem to neglect a few important baselines in the same problem settings. For example,
- RLPD (Ball et al., 2023) uses online RL directly and learn from scratch using a combination of offline data and online data.
- CalQL (Nakamoto et al, 2024) improves CQL so that it fine-tunes better.
For offline RL methods, ReBRAC [1] also experimented with offline to online fine-tuning with strong performance.

*Some results are missing confidence intervals.*
- Table 2 and Figure 2 do not have confidence intervals, which make it hard to tell how significant the results are.

[1] Tarasov, Denis, et al. "Revisiting the minimalist approach to offline reinforcement learning." Advances in Neural Information Processing Systems 36 (2024).

**Questions:**

- Figure 1: What is the difference between offline model and BC model? Is offline model trained with offline RL?

---

### Official Review · Reviewer_3ksT · 2024-11-04

**Soundness:** 3
**Presentation:** 2
**Contribution:** 2
**Rating:** 5
**Confidence:** 4

**Summary:**

The paper introduces a novel framework called Behavior Adaption Q-Learning (BAQ). BAQ aims to address the challenges of transitioning reinforcement learning models from static offline training environments to dynamic online settings. The framework utilizes an implicit behavioral model to imitate and adapt behaviors from offline datasets, enhancing the model's ability to handle out-of-distribution state-action pairs during online deployment. BAQ employs a composite loss function that mimics the offline data-driven policy while dynamically adjusting to new online experiences. The approach includes behavior cloning integration, dynamic Q-value adjustment, and priority-based sample rebalancing. Extensive empirical evaluations demonstrate that BAQ outperforms existing methods, offering improved adaptability and reduced performance degradation across various RL settings.

**Strengths:**

The proposed BAQ approach is clear and simple. The integration of behavior cloning and dynamic Q-value adjustments is straightforward yet effective, making the framework easy to understand and implement.

**Weaknesses:**

- A significant drawback of the paper is the omission of important baselines such as RLPD [1] and BOORL [2], among many others. Including these baselines would have provided a more comprehensive comparison and a better understanding of how BAQ compares against state-of-the-art methods.

- The paper has some issues in writing and formatting. For example, there is a misuse between \citet and \citep starting from Line 348.

- The proposed method only applies to CQL and IQL. It is unclear how the proposed method applies to general offline RL methods.


References

[1] Ball, Philip J., et al. "Efficient online reinforcement learning with offline data." International Conference on Machine Learning. PMLR, 2023.

[2] Hu, Hao, et al. "Bayesian Design Principles for Offline-to-Online Reinforcement Learning." arXiv preprint arXiv:2405.20984 (2024).

**Questions:**

See the Weakness section.

---

### Official Review · Reviewer_p616 · 2024-11-08

**Soundness:** 2
**Presentation:** 2
**Contribution:** 1
**Rating:** 3
**Confidence:** 3

**Summary:**

This paper proposes a weighting scheme for offline-to-online RL, which depress the impact of out-of-distribution (OOD) actions in Q-value estimation while assigning higher sampling priority to OOD actions from the buffer during the online phase. However, the experimental results on three D4RL tasks demonstrate only marginal improvement.

**Strengths:**

This kind of method is a general plug-in and easy to combine with various Q value based methods.  The authors also provide practical implementation for both CQL and IQL.

**Weaknesses:**

The logic of this paper is difficult to follow, and it is unclear how BAQ addresses the offline-to-online RL issue.

From the experimental results, BAQ combined with CQL or IQL does not outperform the original CQL. While the authors emphasize the total normalized score, this metric is less meaningful since CQL outperforms BAQ in 7 out of 9 tasks. Many claims in the paper lack sufficient evidence and clarity. For example, the statement in Line 375, *"Overall, our approach not only surpasses all baseline methods in total scores for both IQL and CQL but also demonstrates strong stability and generalization across diverse tasks,"* and the claim in Line 398, *"In contrast, our method has a robust performance from the beginning without relying on such additional mechanisms,"* appear overstated or even incorrect based on the experimental results.



Some implementation details are unclear. For instance, the value of $\alpha$ in Eq. (10) is not specified in the main text or appendix (though it seems to be $\alpha = 0.6$ based on the code). Additionally, details regarding the online fine-tuning of CQL and IQL are missing.



The hyperparameters $k_q$ and $k_\rho$ are manually set for each task, but there is no discussion on whether these hyperparameters are sensitive or how they impact the results.



Minor:

Repetition: Line 252, "both offline and online interactions"

**Questions:**

1. Why should the total normalized score be considered comparable? It is evident that CQL achieves the highest scores in 7 out of 9 tasks.

2. Why does the performance comparison in Table 1 focus solely on the initial 30,000 steps? Does the evaluation in Table 1 use the best policy or the final policy within the first 30,000 steps?
3. What defines the "early stage of training" in Figure 2? Furthermore, why does the figure omit the results for CQL?

4. Does online fine-tuning of CQL/IQL reduce to the case where $w(s,a)=1$ and $\rho_i=1$ for all offline and online samples?

5. In Figure 2, is CQL or IQL used as the backbone of BAQ? How is the claim *"Our BAQ consistently leads in most tasks with higher normalized scores early in the fine-tuning process"* justified from Figure 2? BAQ appears to perform competitively only in two HalfCheetah tasks and Hopper-medium-replay.

---

### Official Review · Reviewer_ipZK · 2024-11-11

**Soundness:** 2
**Presentation:** 3
**Contribution:** 2
**Rating:** 3
**Confidence:** 3

**Summary:**

The paper proposes BAQ, a new technique to handle out-of-distribution state-action pairs in offline-to-online reinforcement learning. BAQ adds two new components on top of standard offline RL algorithms (CQL, IQL) during fine-tuning - a replay buffer consisting of offline and online data with priority sampling, and a weighted Q-learning mechanism.

BAQ modifies the TD-error objective by giving higher weightage to transitions where the chosen action is similar to that of a BC policy trained on the offline dataset. On the other hand, the priority sampling mechanism replays out-of-distribution actions with higher probability during the TD-updates.

The method is evaluated on MuJoCo environments from the D4RL family and is shown to perform better than baselines such as CQL/IQL + SO2/SUF on several tasks during initial fine-tuning epochs.

**Strengths:**

1. The paper is well written and easy to follow. The idea is novel to the best of my knowledge and placed well within the context of existing literature.

2. The ablation studies are helpful and show that both priority sampling and weighted Q-learning are important for BAQ.

**Weaknesses:**

The evaluations are weak
1. The algorithm is evaluated only on MuJoCo environments from D4RL. The authors should also evaluate the algorithm on harder domains such as AntMaze, Franka-Kitchen and Adroit.

2. Even on MuJoCo, the reported performances are those after 30,000 online fine-tuning steps, and not after running the online fine-tuning to completion. As such, the numbers reported here seem significantly lower than those of running existing fine-tuning methods like Cal-QL to completion. For eg. on the medium environments, Cal-QL reaches normalized score of ~ 100 after online fine-tuning (see Table 5 - https://arxiv.org/pdf/2303.05479) whereas the numbers reported in Table 1 and Table 2 for BAQ are significantly lower.

This makes it hard to compare BAQ against existing online fine-tuning algorithms.

3. Comparisons against fine-tuning methods like Cal-QL are missing, and it is unclear how BAQ helps with the distribution shift problem in offline-to-online RL.

**Questions:**

1. What is the performance on MuJoCo after running online fine-tuning using BAQ to completion? How does BAQ perform in harder environments like AntMaze, Franka-Kitchen and Adroit?

2. Would adding the priority sampling and weighted Q-learning mechanisms enhance the performance of off-the-shelf fine-tuning methods (for eg. Cal-QL)?

3. It seems that the weighting Q-learning mechanism reduces the contribution of an OOD update, whereas the priority sampling mechanism increases it (in expectation). It is unclear how doing both simultaneously is helping with the distribution shift problem. Can the authors explain it in more detail?

---

### Meta-Review · Area_Chair_XwoZ · 2024-12-22

**Metareview:**

The paper proposes BAQ, a new technique to handle out-of-distribution state-action pairs in offline-to-online reinforcement learning. BAQ adds two new components on top of standard offline RL algorithms (CQL, IQL) during fine-tuning - a replay buffer consisting of offline and online data with priority sampling, and a weighted Q-learning mechanism.

Several concerns were raised by the reviewers, but no rebuttal was provided. Neither was any concern resolved. Unfortunately, we cannot accept the paper as a result.

**Additional Comments On Reviewer Discussion:**

There was no discussion, though the reviewers' strengths and weaknesses adequately quantifies the main points that I request the authors to take a look at.

---

### Decision · Program_Chairs · 2025-01-22

Reject